# Beyond Traditional Transfer Learning: Co-finetuning for Action Localisation

## Abstract

Transfer learning is the predominant paradigm for training deep networks on small target datasets. Models are typically pretrained on large "upstream" datasets for classification, as such labels are easy to collect, and then finetuned on "downstream" tasks such as action localisation, which are smaller due to their finer-grained annotations.

In this paper, we question this approach, and propose co-finetuning – simultaneously training a single model on multiple "upstream" and "downstream" tasks. We demonstrate that co-finetuning outperforms traditional transfer learning when using the same total amount of data, and also show how we can easily extend our approach to multiple "upstream" datasets to further improve performance. In particular, co-finetuning significantly improves the performance on rare classes in our downstream task, as it has a regularising effect, and enables the network to learn feature representations that transfer between different datasets. Finally, we observe how co-finetuning with public, video classification datasets, we are able to achieve significant improvements for spatio-temporal action localisation on the challenging AVA and AVA-Kinetics datasets, outperforming recent works which develop intricate models.

## 1 Introduction

The computer vision community has made impressive progress in video classification with deep learning, first with Convolutional Neural Networks (CNNs) (Krizhevsky et al., 2012; Carreira & Zisserman, 2017; Feichtenhofer et al., 2019), and more recently with transformers (Vaswani et al., 2017; Arnab et al., 2021a; Fan et al., 2021). However, progress in other more challenging video understanding tasks, such as spatio-temporal action localisation (Pan et al., 2021; Tang et al., 2020; Zhao et al., 2022), has lagged behind significantly in comparison.

One major reason for this situation is the lack of data with fine-grained annotations which are not available for such complex tasks. To cope with this challenge, the de facto approach adopted the state-of-the-art is transfer learning (popularised by Girshick et al. (2014)). In the conventional setting, a model is first pre-trained on a large "upstream" dataset, which is typically labelled with classification annotations as they are less expensive to collect. The model is then "finetuned" on a smaller dataset, often for a different task, where fewer labelled examples are available (Mensink et al., 2022). The intuition is that a model pre-trained on an auxiliary, "upstream" dataset learns generalisable features, and therefore its parameters do not need to be significantly updated during finetuning. For video understanding, the most common "upstream" dataset is Kinetics (Kay et al., 2017), demonstrated by the fact that the majority of recent work addressing the task of spatio-temporal action localisation pretrain on it (Zhao et al., 2022; Pan et al., 2021; Wu et al., 2019). Similarly for image-level tasks, ImageNet (Deng et al., 2009) is the most common "upstream" dataset.

Our objective in this paper is to train more accurate models for spatio-temporal action detection, and we do so by proposing an alternate training strategy of co-finetuning. Instead of using the additional classification data in a separate pre-training phase, we simultaneously train for both classification and detection tasks. Intuitively, the additional co-finetuning datasets can act as a regulariser during training, benefiting in particular the rare classes in the target dataset which the network could otherwise overfit on. Moreover,

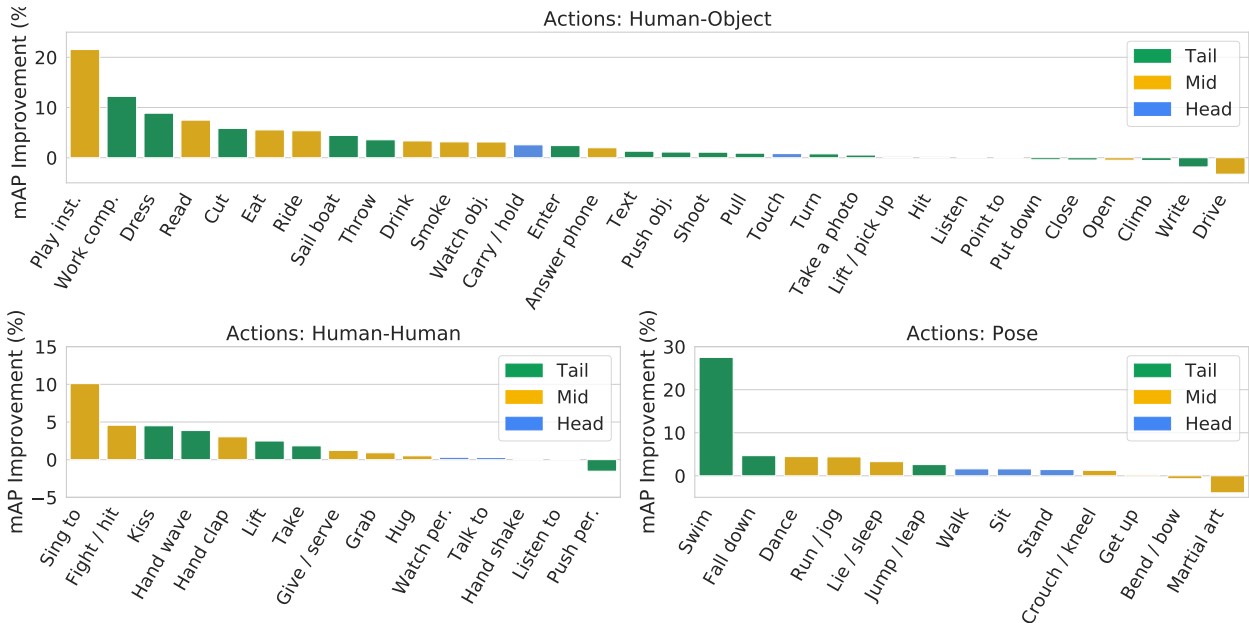

Figure 1: Improvements achieved by our co-finetuning strategy on the AVA dataset Gu et al. (2018). AVA is a long-tailed dataset, and we attain significant improvements, particularly on the rare classes (shown by the "Tail" and "Mid" subsets) in the dataset. Our improvements are also consistent across the three action types defined in the AVA dataset. We split the 60 class labels from AVA into "Head" classes ($> 10,000$ ground truth instances), "Tail" classes ($< 1000$ instances), and the remaining ones into "Mid" classes (detailed in Sec. 4.2).

discriminative features which are learned for classification datasets may also transfer to the detection dataset, even though the target task and labels are different.

Our thorough experimental analyses confirm these intuitions. As shown by Fig. 1, our co-finetuning strategy improves spatio-temporal action localisation results for the vast majority of the action classes on the AVA dataset, improving substantially on the rarer classes with few labelled examples. In particular, co-finetuning performs better than traditional transfer learning when using the same total amount of data. Moreover, with co-finetuning we can easily make use of additional "upstream" classification datasets during co-finetuning to improve results even further.

Our approach is thus in stark contrast to previous works on spatio-temporal action detection which develop complex architectures to model long-range relationships using graph neural networks (Arnab et al., 2021b; Wang & Gupta, 2018; Sun et al., 2018; Baradel et al., 2018; Zhang et al., 2019), external memories (Wu et al., 2019; Pan et al., 2021; Tang et al., 2020) and additional object detection proposals (Wang & Gupta, 2018; Wu & Krahenbuhl, 2021; Tang et al., 2020; Zhang et al., 2019). Instead, we use a simple detection architecture and modify only the training strategy to achieve higher accuracies, outperforming prior work.

We conduct thorough ablation analyses to validate our method, and make other findings too, such as the fact that although Kinetics (Kay et al., 2017) is the most common upstream pretraining dataset for video tasks, and used by all previous work addressing spatio-temporal action detection on AVA, Moments in Time (Monfort et al., 2019) is actually better. Finally, a by-product of our strategy of co-finetuning with classification tasks is that our network can simultaneously perform these tasks, and is competitive with the state-of-the-art on these datasets too.

## 2 Related Work

We first discuss transfer learning, multi-task learning and domain adaptation, which our proposed co-finetuning approach is related to. We then review action detection models, as this is our final task of interest.

**Transfer- and multi-task learning** The predominant paradigm for training a deep neural network is to pre-train it on a large "upstream" dataset with many annotations, and then to finetune it on the final "downstream" dataset of interest. This approach was notably employed by R-CNN (Girshick et al., 2014) which leveraged pre-trained image classification models on ImageNet (Deng et al., 2009), for object detection on the markedly smaller Pascal VOC dataset (Everingham et al., 2015). Since then, this strategy has become ubiquitous in training deep neural networks (which require large amounts of data to fit) across a wide range of vision tasks (Mensink et al., 2022). The training strategy of Girshick et al. (2014) is a form of "inductive transfer learning" based on the taxonomy of Pan & Yang (2009), and we simply refer to it as "traditional transfer learning" in this paper.

We further note that it is common to perform multiple stages of transfer, in which case the "downstream" task can become the "upstream" task for the next stage of training (Mensink et al., 2022). For example, video classification models are often initialised from image models pretrained on ImageNet-1K or ImageNet-21K (Deng et al., 2009), and then finetuned on Kinetics (Wang et al., 2016; Carreira & Zisserman, 2017; Arnab et al., 2021a; Bertasius et al., 2021; Liu et al., 2022). These video classification models are then again finetuned on the final task of interest, such as spatio-temporal action detection on AVA (Girdhar et al., 2019; Wang et al., 2018; Zhao et al., 2022). In this work, we focus on co-finetuning video classification and action detection datasets together.

Our co-finetuning strategy, of jointly training a model on multiple upstream and downstream datasets simultaneously, is closely related to multi-task learning. Multi-task learning aims to develop models that can address multiple tasks whilst sharing parameters and computation among them (Caruana, 1997). However, in computer vision, the focus of many prior works has been to predict multiple outputs (for example semantic segmentation and surface normals) (Eigen & Fergus, 2015; Kokkinos, 2017; Kendall et al., 2018; Kurin et al., 2022) given a single input image. These works have typically observed that although these multi-task models are more versatile, their accuracies are lower than single-task models. Moreover, this accuracy deficit worsens as the number of tasks grows, or if multiple unrelated tasks are simultaneously performed (Kokkinos, 2017; Zamir et al., 2018; Sener & Koltun, 2018). Training a model to perform multiple tasks on the same input often results in one task dominating the other, and numerous strategies have been proposed to mitigate this problem by adapting the losses or gradients from each task (Sener & Koltun, 2018; Chen et al., 2020b).

Our scenario differs, in that although our network is capable of performing multiple tasks, it only performs a single task at a time for a given input. This is also the standard setting of multi-task learning in natural language processing (Collobert & Weston, 2008; Raffel et al., 2019), though not in vision, with Maninis et al. (2019) referring to it as "single tasking of multiple tasks."

Our decision of performing a single task at a time ensures that our training strategy is simple, and does not require additional hyperparameters to stabilise, like previous multi-task learning methods in computer vision. Moreover, we are primarily interested in improving performance on a target task, and using the additional datasets to learn more generalisable features and for regularisation.

**Domain adaptation** Our proposed approach is also related to domain adaptation, another form of transfer learning, which aims to exploit labelled data from a "source" domain to perform better on a "target" domain which has limited or few annotations (Csurka, 2017; Tzeng et al., 2017). Extensions to multiple source domains, known as multi-source domain adaptation (Zhao et al., 2020), have also been proposed. Domain adaptation approaches, however, typically assume that the same task and label space is used between the "source" and the "target" (Csurka, 2017; Koh et al., 2021). Therefore, common applications of domain adaptations include training on synthetic segmentation datasets and evaluating on real-world datasets with the same label space (Peng et al., 2017), training and evaluating across different camera traps for wildlife monitoring (Beery et al., 2018), or across time and geographical locations in satellite imaging and mapping (Jean et al., 2016).

Our work differs in that we use different tasks and label spaces for our "upstream" and "downstream" tasks. In particular, our upstream tasks consist of video classification, and downstream tasks are spatio-temporal action localisation, where there are no labels in common between upstream and downstream tasks. Moreover,

we show how we can leverage our training strategy to outperform existing methods which have developed intricate models to solve this task.

**Spatio-temporal action detection models** Current state-of-the-art action detection models (Pan et al., 2021; Fan et al., 2021; Tang et al., 2020) are based on the Fast R-CNN architecture (Girshick, 2015), and use external person detections as region proposals which are pooled and classified. We note, however, that some recent works (Chen et al., 2021; Zhao et al., 2022) build off DETR (Carion et al., 2020) and do not require external proposals, although they are not as performant.

Localising actions in space and time often requires capturing long-range contextual information, and state-of-the-art approaches have explicitly modelled this context to improve results. A common theme is to use graph neural networks to explicitly model relationships between actors, objects and the scene (Arnab et al., 2021b; Baradel et al., 2018; Sun et al., 2018; Wang & Gupta, 2018). Many approaches also use off-the-shelf object detectors to locate relevant scene context (Baradel et al., 2018; Herzig et al., 2021; Wang & Gupta, 2018; Zhang et al., 2019). To model long-range temporal contexts spanning several minutes, (Wu et al., 2019; Tang et al., 2020) have proposed external memory banks which are populated by first running a video feature extractor over the input video, and are thus not trained end-to-end. The most accurate methods have included a combination of external memories and spatio-temporal graphs (Pan et al., 2021; Tang et al., 2020), and are thus intricate models which require complex training and inference procedures due to the external memory bank.

Our proposed approach, in stark contrast, uses a simple Fast R-CNN architecture, without any explicit graph modelling (Arnab et al., 2021b; Zhang et al., 2019; Sun et al., 2018), auxiliary object detections (Herzig et al., 2021; Wang et al., 2020; Zhang et al., 2019) or external memories (Wu et al., 2019; Pan et al., 2021; Tang et al., 2020), and achieves improvements over the baseline model that are competitive or even better than the complex models of (Arnab et al., 2021b; Pan et al., 2021; Sun et al., 2018; Wu et al., 2019), by simply changing the training strategy.

## 3 Proposed Approach

Our objective is to train a model for spatio-temporal detection, using additional video classification data to improve our model's performance. We first describe our model in Sec. 3.1, before detailing how we co-finetune it using additional classification data in Sec. 3.2. Finally, we discuss our training strategy in Sec. 3.3.

### 3.1 Model

Our main objective is to study co-finetuning strategies for training action detection models. As such, we choose a transformer encoder model with separate heads for each classification and detection dataset that we co-finetune with (Fig. 2).

Given an input $\mathbf{x} \in \mathbb{R}^{T \times H \times W \times C}$, we first extract $N$ tokens from the input video, $\mathbf{z}^0 = \{z_1, z_2, \ldots, z_N\}$, with $\mathbf{z} \in \mathbb{R}^{t \times h \times w \times d}$. Here, $d$ denotes the hidden dimension of the model, and the spatio-temporal dimensions of the tokens, $(t, h, w)$, depend on the tubelet size (Arnab et al., 2021a) when tokenizing the input. These tokens are then processed by a transformer encoder consisting of $L$ layers to result in $\mathbf{z}^L$. Depending on the task (video classification, or action detection), the encoded tokens, $\mathbf{z}^L$, are forwarded through a dataset-specific head, as shown in Fig. 2.

**Detection head** As shown in Fig. 2, our detection head follows the Fast R-CNN meta-architecture (Girshick, 2015). We adopt this method as current and previous state-of-the-art approaches (Pan et al., 2021; Feichtenhofer et al., 2019; Wu et al., 2019; Fan et al., 2021) have used it. Specifically, we use person detections, $\mathbf{p} \in \mathbb{R}^{N_p \times 4}$, obtained from an off-the-shelf person detector, to spatio-temporally pool the encoded tokens, $\mathbf{z}^L$, using ROI-Align (He et al., 2017). Here, $N_p$ denotes the number of people detected for the particular input video. We then obtain our predicted logits for the $j^{th}$ person-proposal as $\hat{\mathbf{y}}_j$ using a linear projection. This is described by

$$\mathbf{z}_j^p = \text{ROI-Align}(\mathbf{z}^L, p_j), \qquad \hat{\mathbf{y}}_j = \mathbf{W}\mathbf{z}_j^p. \qquad (1)$$

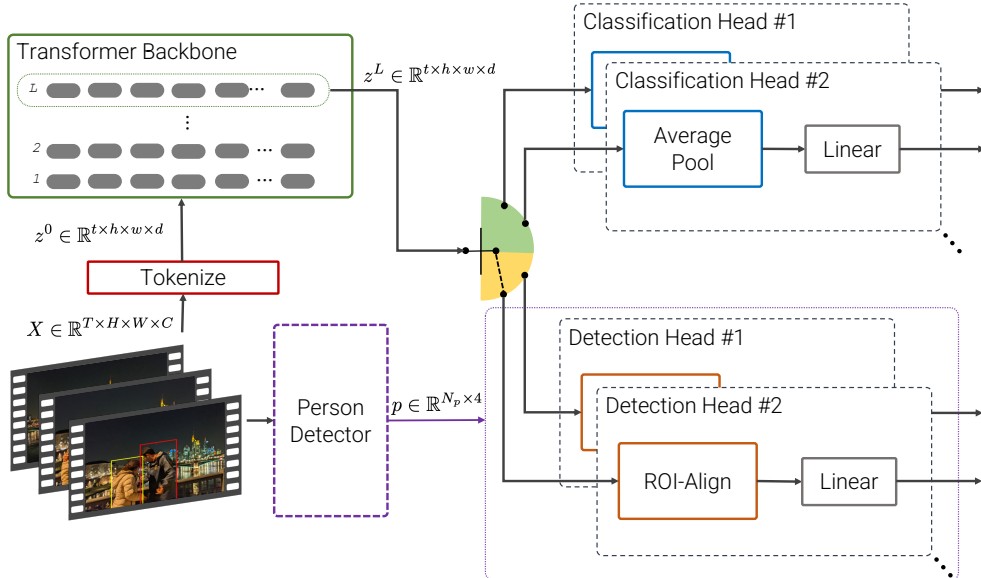

Figure 2: Model overview. A transformer backbone is used to extract spatio-temporal features which are then processed by task-specific heads. Note that we co-finetune on multiple tasks simultaneously.

Note that our person detections, $\mathbf{p}$, are for the central keyframe of the video clip following (Pan et al., 2021; Feichtenhofer et al., 2019; Wu et al., 2019; Fan et al., 2021). For ROI-pooling, we extend the bounding box temporally to form a "tube" following prior work. In practice, our proposals, $\mathbf{p}$, are parameterised as the upper-left and bottom-right co-ordinates of the bounding box, and we can readily adopt other parameterisations as well.

**Classification head**   The video classification head is a simple linear classifier. We extract a representation of the video by spatio-temporally averaging all the encoded tokens. Therefore, $\mathbf{z}^v = \mathrm{Average}(\mathbf{z}^L) \in \mathbb{R}^d$, and the logits predicted for each class for the $i^{th}$ dataset are $\hat{\mathbf{y}} = \mathbf{W}^i \mathbf{z}^v$.

## 3.2   Co-finetuning Strategy

The de facto approach for training deep architectures for computer vision tasks is to pre-train the model on a large "upstream" dataset, such as ImageNet (Deng et al., 2009), and then to finetune it on the smaller target dataset. Each training procedure (pre-training or finetuning), is denoted by

$$\arg\min_{\theta} \mathbb{E}_{(x,y)\in\mathcal{D}} L(f(\mathbf{x};\theta),\mathbf{y}), \tag{2}$$

where $\mathcal{D}$ denotes the dataset consisting of pairs of videos, $\mathbf{x}$ and ground truth labels, $\mathbf{y}$, $L$ is the loss function, and $f$ is a neural network with model parameters, $\theta$. If we denote $\theta = \{\theta_b, \theta_h\}$ as the parameters of the backbone and head respectively, "pre-training" can be understood as optimising all model parameters from random initialisation, whereas "finetuning" initialises the backbone model parameters, $\theta_b$ from pre-training, and trains new head parameters, $\theta_h$, for the final task of interest. Initialisation of network parameters $\theta$ is critical, as it is a non-convex optimisation problem.

Note that it is common, particularly in video understanding, to include multiple stages of pre-training. For example, many video models are initialised from networks pretrained on ImageNet, then finetuned on Kinetics and then finally finetuned for spatio-temporal detection (Girdhar et al., 2019; Köpüklü et al., 2019; Wu et al., 2019; Zhao et al., 2022).

In our proposed approach, we simultaneously finetune our model on multiple datasets: This includes our target dataset of interest, as well as other datasets from which our model can learn expressive features, and which may help to regularise the model. We can express this as

$$\arg\min_{\theta} \mathbb{E}_{(x,y)\in\mathcal{D}_1\cup\mathcal{D}_2\dots\cup\mathcal{D}_{N_d}} L(f(\mathbf{x};\theta),\mathbf{y}), \tag{3}$$

where $N_d$ denotes the total number of datasets. Our model parameters are now $\theta = \{\theta_b, \theta_{h_1}, \theta_{h_2}, \ldots, \theta_{h_{N_d}}\}$ as we have separate head parameters, $\theta_{h_i}$, for each co-finetuning dataset.

### 3.2.1 Implementation

To co-finetune on multiple datasets simultaneously as in Eq. 3, we construct each training minibatch by sampling examples from one particular dataset. We then perform a parameter-update using SGD (or other first-order optimisers) with gradients computed from this batch, before sampling another dataset and iterating. We consider the following two alternatives for sampling minibatches:

**Alternating** Given $N_d$ datasets, we sample batches from datasets sequentially. For example, we first sample a batch from dataset $\mathcal{D}_1$, then $\mathcal{D}_2, \ldots \mathcal{D}_{N_d}$ before continuing from $\mathcal{D}_1$ again.

**Weighted sampling** We sample batches from a dataset with a probability proportional to the number of examples in the dataset. This corresponds to concatenating several datasets together, and drawing batches randomly. It is also possible to set the sampling weights for each dataset, but we avoid this approach to avoid introducing additional hyperparameters to tune.

### 3.2.2 Discussion

Note that it is also possible to construct a minibatch using a mixture of examples from each dataset. However, we found this approach to be less computationally efficient in initial experiments and did not pursue it further. Finally, if we constructed our batches such that we first sampled only examples from dataset $\mathcal{D}_1$, then $\mathcal{D}_2$ and so on, this would be equivalent to finetuning on each dataset sequentially, with the difference that the optimiser state (such as first- and second-order moments of the gradient (Qian, 1999; Kingma & Ba, 2015)) would be carried over from one dataset to the next, which is not typically done when finetuning.

### 3.3 Intuition behind co-finetuning

In this section, we explain our intuitions behind why co-finetuning could be effective, given that by sequentially finetuning a model on $N_d$ different datasets, the model would have access to the same total amount of data.

By co-finetuning on multiple datasets simultaneously, it is possible for the model to learn general, discriminative visual patterns on one dataset that transfer to the other. And this effect may be more pronounced when the target dataset (such as AVA (Gu et al., 2018) for spatio-temporal action detection) is small and contains many tail classes with few labelled examples. Sequentially finetuning a model on multiple datasets may not have the same property due to "catastrophic forgetting" (French, 1999) – the propensity of neural networks to degrade at previous tasks when finetuned on a new one.

Another reason is that deep neural networks, and especially transformers, are prone to overfitting, as evidenced by the numerous regularisation techniques that are typically employed when training them (Touvron et al., 2020; Arnab et al., 2021a; Bello et al., 2021; Wightman et al., 2021). Training with additional data, even if the additional co-finetuning datasets are not for the task of interest, can therefore remediate this problem by acting as a regulariser. In particular, we would expect the benefits to be the most apparent on small and/or imbalanced datasets (since the network may otherwise easily memorise the classes with few examples). Datasets for more complex video understanding tasks, such as action detection, are typically small due to their annotation cost. We now validate these hypotheses experimentally in the next section.

## 4 Experiments

### 4.1 Experimental Setup

**Datasets** *AVA* (Gu et al., 2018) is the largest dataset for spatio-temporal action detection, consisting of 430, 15-minute video clips obtained from movies. The dataset consists of 80 atomic actions, of which 60 are

used for evaluation. These actions are annotated for all actors in the video, where one person is typically simultaneously performing multiple actions. The dataset annotates keyframes at every second in the video, and following standard practice, we report the Frame AP at an IoU threshold of 0.5 using the annotations from version 2.2 (Gu et al., 2018).

*Kinetics* (Kay et al., 2017) is a collection of clip-classification datasets consisting of 10 second video clips focusing on human actions. The Kinetics datasets have grown progressively, with Kinetics 400, 600 and 700, annotated with 400, 600 and 700 action classes respectively. The dataset sizes range from 240,000 (Kinetics 400) to 530,000 (Kinetics 700). To avoid confusion with AVA-Kinetics, described below, we denote these datasets as Kinetics$^{\text{Clip}}$.

*AVA-Kinetics* (Li et al., 2020) adds detection annotations, following the AVA protocol, to a subset of Kinetics 700 videos. Note that only a single keyframe, in a 10-second Kinetics clip, is annotated. To avoid confusion with Kinetics$^{\text{Clip}}$, we use Kinetics$^{\text{Box}}$ to denote the Kinetics videos with detection annotations. Therefore, the training and validation splits of AVA-Kinetics are the union of the respective AVA and Kinetics$^{\text{Box}}$ dataset splits.

*Moments in Time* (Monfort et al., 2019) consists of 800,000 clips, collected from YouTube, each 3-seconds long and annotated with one of the 339 classes.

*Something-Something v2* (Goyal et al., 2017) consists of about 220,000 short video clips annotated with 174 classes showing humans interacting with everyday objects.

**Implementation details** Our transformer backbone architecture is the ViViT Factorised Encoder (Arnab et al., 2021a), with $16 \times 16 \times 2$ tubelets, following the authors' public implementation. We chose this backbone as it is a state-of-the-art video classification model. Note that ViViT, like other popular transformers for video (Bertasius et al., 2021; Liu et al., 2022; Zhang et al., 2021) is initialised from a ViT image model pretrained on ImageNet-21K (Deng et al., 2009).

We use the same person detections as prior work (Wu et al., 2019; Feichtenhofer et al., 2019; Fan et al., 2021; Arnab et al., 2021b). For AVA, these were obtained using a Faster R-CNN (Ren et al., 2015) detector, and released publicly by Wu et al. (2019), achieving an AP of 93.9 for person detection on the AVA validation set. For AVA-Kinetics, these were obtained using CenterNet (Zhou et al., 2019) by Li et al. (2020), obtaining an AP of 77.5 on AVA-Kinetics for person detection. Furthermore, note that we report results from processing a single view for all tasks.

All models are trained on 32 frames with a temporal stride of 2. We train for 15 epochs using synchronous SGD with a momentum of 0.9, where an epoch is defined as the sum of the size of each dataset. Exhaustive details are included in Appendix C, and we will release code upon acceptance.

### 4.2 Ablation Study

We first conduct thorough ablation studies of our co-finetuning approach in Tables 1 through 4. For rapid experimentation, unless otherwise stated, we use a ViViT-Base Factorised Encoder as our backbone model. Videos are resized such that the longest and shortest sides have resolutions of 224 and 140 respectively (denoted as 140p). We use the "Weighted" minibatch sampling strategy (Sec. 3.2) unless otherwise stated.

**Comparison of co-finetuning and traditional finetuning with the same data** Tables 1 and 2 compare the effect of traditional transfer learning, and co-finetuning, with the same total amount of data, showing that co-finetuning achieves superior action detection results on AVA, as well as improved video classification accuracy, in all cases. Note that the total computational cost of co-finetuning is actually identical to training pretraining and finetuning. This is because we train for the same number of epochs on the upstream datasets. Therefore, all experiments (baselines and our method) in Tables1 and 2 have the same training time.

Note that in all experiments, our ViViT backbone model is initialised from an ImageNet-21K pretrained checkpoint, following common practice in training transformer models for video (Arnab et al., 2021a; Berta-

Table 1: Comparison of traditional finetuning, and our proposed co-finetuning, with the same total amount of data. Observe how co-finetuning consistently outperforms traditional finetuning. We report single-view accuracies for all datasets, using a ViViT-Base backbone. "TI" denotes the total number of training iterations, in thousands, using a batch size of 128.

(a) Kinetics 400 as upstream classification dataset

| Pretrain | Finetune | AVA | K400 | TI (k) |
|---|---|---|---|---|
| K400 | – | – | 74.9 | 55.2 |
| K400 | AVA | 23.4 | – | 88.2 |
| – | K400 + AVA | **25.2** | **76.2** | 88.2 |

(b) Moments in Time as upstream dataset

| Pretrain | Finetune | AVA | MiT | TI (k) |
|---|---|---|---|---|
| MiT | – | – | 36.8 | 123.5 |
| MiT | AVA | 24.8 | – | 156.5 |
| – | MiT + AVA | **26.1** | **38.1** | 156.5 |

Table 2: Comparison of traditional finetuning, and our proposed co-finetuning, using two upstream datasets. Co-finetuning consistently outperforms traditional finetuning with the same overall data, even considering that there are multiple orders in which two upstream datasets can be pretrained in with traditional finetuning (K400→MiT denotes first pretraining on K400 and then MiT). The final column shows the total number of training iterations, in thousands, using a batch size of 128.

| Pretrain | Finetune | AVA | K400 | MiT | Total Iterations (k) |
|---|---|---|---|---|---|
| K400 | MiT | – | – | 37.2 | 178.8 |
| MiT | K400 | – | 76.3 | – | 178.8 |
| K400→MiT | AVA | 25.3 | – | – | 211.8 |
| MiT→K400 | AVA | 25.1 | – | – | 211.8 |
| – | MiT + K400 + AVA | **26.3** | **76.7** | **38.9** | 211.8 |

sius et al., 2021; Liu et al., 2022; Yan et al., 2022; Zhang et al., 2021), and we omit this from the "Pretrain" column for clarity.

Table 1(a) shows that co-finetuning on Kinetics 400 and AVA jointly improves the mAP on AVA by 1.8 points, or 7.7% relative, whilst using the same total amount of data. An added advantage of co-finetuning is that the model can perform multiple tasks – unlike traditional transfer learning, the model does not "forget" the upstream dataset (French, 1999) as it is simultaneously optimised to perform both tasks. We can see that single-crop classification performance on Kinetics 400 increases by 1.3 points. Moreover, there is a similar increase for Moments in Time (MiT) in Tab. 1(b).

Table 1(b) shows that co-finetuning on MiT also improves AVA action localisation accuracy, achieving an improvement of 1.3 points. Note that existing work reporting results on AVA (Arnab et al., 2021b; Girdhar et al., 2019; Feichtenhofer et al., 2019; Pan et al., 2021) have used Kinetics as the upstream classification dataset for transfer learning. However, Tab. 1(b) shows that pretraining on MiT outperforms Kinetics 400 by 1.4 points. Note that whilst MiT contains more video clips than Kinetics 400 (800,000 versus 240,000), its clips are also much shorter at 3 seconds compared to the 10 seconds of Kinetics, and hence the total duration of video in both datasets is similar.

Finally, Tab. 2 considers the scenario where we have two upstream classification datasets, Kinetics and MiT. With traditional transfer learning, there are two orders in which we can pretrain on the upstream clip classification datasets: first on Kinetics and then MiT, or vice versa. We observe that co-finetuning outperforms both of these alternatives on AVA action detection, and moreover, does not require the additional training hyperparameter of choosing the order to use the upstream datasets.

**Adding more co-finetuning datasets** Table 3 shows how adding more clip-classification datasets for co-finetuning improves our spatio-temporal action detection performance on AVA. We observe progressive improvements from adding Kinetics, Moments in Time (MiT) and Something-Something (SSv2) respectively, although the gains diminish with additional datasets. We chose these clip classification datasets for co-finetuning as they are the largest available public datasets that we are aware of. Note that Kinetics and

Table 3: By using more clip-classification datasets for co-training, spatio-temporal action detection performance, measured by the mAP on AVA, gradually increases. Note how the improvement is primarily in the rare classes, denoted by the "Mid" and "Tail" categories. Refer to Sec. 4.2 and Fig. 1 for additional details on how these splits were constructed.

| Training / Co-training datasets | Head | Mid | Tail | Overall |
|---|---|---|---|---|
| K400→AVA traditional finetuning baseline | 65.5 | 29.7 | 6.9 | 23.4 |
| K400 + AVA | 65.6 | 31.8 | 8.9 | 25.2 |
| K400 + MiT + AVA | 66.1 | 33.7 | 9.5 | 26.3 |
| K400 + MiT + SSv2 + AVA | $\mathbf{66.7}_{+1.2}$ | $\mathbf{34.1}_{+4.4}$ | $\mathbf{10.1}_{+3.2}$ | $\mathbf{26.8}_{+3.4}$ |

Table 4: Effect of minibatch sampling strategies. We co-finetune on K400, MiT and AVA simultaneously.

| | AVA | K400 | MiT |
|---|---|---|---|
| Alternating | 25.0 | 76.4 | 34.8 |
| Weighted sampling | **26.3** | **76.7** | **38.9** |

Moments in Time consist of videos that are similar in domain to AVA as they are collected from YouTube, and feature people. SSv2 (Goyal et al., 2017), on the other hand, consists of videos from a different domain, consisting mostly of objects being manipulated, and requires a model to capture fine-grained motion patterns. Nevertheless, we observe noticeable improvements from using SSv2, suggesting that the additional data has a regularising effect on the model, and that motion cues learned from SSv2 are useful for AVA detection.

**What classes does co-finetuning benefit?** A hypothesis for co-finetuning was that it enables a model to learn discriminative visual patterns on one dataset that transfer to the other, and that this would particularly benefit rare classes in the target dataset that the model with otherwise overfit on.

To validate this hypothesis, we split the 60 class labels from AVA into "Head", "Mid" and "Tail" classes. "Head" classes are defined as those with more than 10,000 labelled ground truth examples in the training set, whereas "Tail" classes have less than 1000 instances. Classes which don't fall into either of these categories are defined as "Mid" classes. There are 8 "Head", 23 "Mid" and 29 "Tail" classes respectively, as can be seen in Fig. 1 and Tab. 10 of the appendix.

As shown in Tab. 3, adding more co-finetuning datasets improves our AVA detection mAP for "Mid" and "Tail" classes. In particular, our best co-finetuned model improves "Mid" classes by 4.4 points, or 14.8% relative. The relative improvement for "Tail" classes is even more, at 46.4%. Figure 1 details this further, showing the improvement from co-finetuning on each action class.

**Improvements from training strategy compared to modelling** Table 3 shows that by leveraging public clip classification datasets, we are able to increase performance on AVA, measured by the mAP, by 3.4 points. To put this in context, we notice that the spatio-temporal actor-object graph proposed by Arnab et al. (2021b) improved the author's baseline by 2.2 points. The external memory of Wu et al. (2019) which requires precomputing video features for the entire video offline before training and inference obtains an improvement over the baseline of 2.9 points. The method of Pan et al. (2021), which combines external memories and spatio-temporal graphs, improves over its respective baseline by 3.4 points, but is significantly more complex. Therefore, we conclude that co-finetuning on public clip-classification datasets provides a simple method of increasing action detection models by amounts commensurate with, or even higher than, the latest state-of-the-art methods. We caution, however, that definitive comparisons are difficult to make as each of the aforementioned methods uses slightly different baselines, backbones and code bases.

**Minibatch sampling strategy for co-finetuning** Table 4 evaluates the effect of the two different minibatch sampling strategies described in Sec. 3.2. The "alternating" minibatch strategy samples an equal

Table 5: Comparison to the state-of-the-art (reported with mean Average Precision) on AVA (Gu et al., 2018) and AVA-Kinetics Li et al. (2020). For AVA, we use the latest v2.2 annotations. Previous work followed the traditional finetuning approach of pretraining on Kinetics and then finetuning on AVA. Note that separate models are trained for AVA and AVA-Kinetics.

|  | Pretrain | Views | AVA | AVA-Kinetics |
|---|---|---|---|---|
| Action Transformer (Li et al., 2020) | K400 | 1 | – | 23.0 |
| MViT-B (Fan et al., 2021) | K400 | 1 | 27.3 | – |
| Unified (Arnab et al., 2021b) | K400 | 6 | 27.7 | – |
| WOO (Chen et al., 2021) | K600 | 1 | 28.3 | – |
| MViT-B (Fan et al., 2021) | K600 | 1 | 28.7 | – |
| SlowFast R101 (Feichtenhofer et al., 2019) | K600 | 6 | 29.8 | – |
| ACAR (Pan et al., 2021) | K600 | 1 | 31.4 | – |
| AIA (Tang et al., 2020) | K700 | 18 | 32.3 | – |
| ACAR (Pan et al., 2021) | K700 | 1 | 33.3 | 35.8 |
| TubeR (Zhao et al., 2022) | IG→K400 | 2 | 33.6 | – |
| ViViT-L (K700+MiT+SSv2) | – | 1 | 32.8 | 33.1 |
| ViViT-L (K700+MiT+SSv2) | WTS | 1 | **36.1** | **36.2** |

number of minibatches from each of the datasets during co-finetuning, regardless of the size of the dataset. Consequently, the accuracy on the largest dataset (MiT) decreases as not enough iterations of backpropagation are applied to it. On the other hand, the accuracy on the smallest dataset (AVA) decreases as too many optimisation iterations are performed on it, and the network begins to overfit on it. The "Weighted sampling" strategy does not suffer these problems, and performs better overall, as batches are drawn from the co-finetuning datasets in proportion to the size of each dataset. As a result, we use the "Weighted sampling" strategy in all of our experiments.

**Leveraging larger-scale pretraining and resolutions** Using higher resolutions as well as backbone models pretrained on larger datasets is a common method of improving performance. Tables 8 and 9 of the appendix shows that we continue to achieve improvements from co-finetuning in both of these scenarios. For large-scale pretraining, we pretrained our ViVIT backbone on the larger Weak Textual Supervision (WTS) dataset of Stroud et al. (2020) which consists of about 60 million weakly-labelled, web-scraped videos.

### 4.3 Comparison to the state-of-the-art

**AVA** We compare to the existing state-of-the-art approaches on AVA in Tab. 5. All methods, excluding TubeR (Zhao et al., 2022) and WOO (Chen et al., 2021) use external person detections as inputs to the model. We use the same person detections as Fan et al. (2021); Feichtenhofer et al. (2019); Arnab et al. (2021b); Wu et al. (2019) as aforementioned. For state-of-the-art comparisons, our model uses frames with a resolution of 320 on the shorter side. Similar to object detection in images (Lin et al., 2017; Singh & Davis, 2018), we find that increasing the spatial resolution improves results, as detailed in Appendix B. We perform inference on a single view, noting that prior work often averages results over multiple resolutions, and left/right flips (Feichtenhofer et al., 2019; Arnab et al., 2021b; Tang et al., 2020; Zhao et al., 2022).

Table 5 shows that by co-finetuning on just publicly available classification datasets, we can achieve an AP of 32.8. The previous best supervised-learning result on AVA, was achieved by TubeR Zhao et al. (2022) which used pretraining on the Instagram (IG) dataset (Ghadiyaram et al., 2019) containing 65 million videos. When we pre-train our initial model with WTS (Stroud et al., 2020) (which consists of 60 million videos, and thus of a similar scale to IG), we achieve a mean AP of 36.1.

**AVA-Kinetics** The final column of Tab. 5 compares to previous methods on AVA-Kinetics. We use the same person detections as Li et al. (2020), which obtain an mAP for person detection of 77.5 on the

Kinetics$^{\text{Box}}$ split of this dataset. ACAR (Pan et al., 2021; Chen et al., 2020a), in contrast, uses a better person detector which achieves an mAP of 84.4, but has not been released.

We can separate the effect of person detections by evaluating using ground truth bounding boxes instead. In this case, our model without WTS-pretraining achieves an mAP of 47.0 on both Kinetics$^{\text{Box}}$ and the overall dataset. ACAR is 3.4 points lower using ground truth boxes on Kinetics$^{\text{Box}}$, attaining 43.6 (the authors do not report the overall dataset mAP in this setting).

With WTS-pretraining, our co-finetuned model achieves, to our knowledge, the best known result on this dataset with an mAP of 36.2.

## 5 Conclusion and Future Work

Recent advances in spatio-temporal action localisation have been driven by developing more complex models which use external memories to capture long-term temporal context (Wu et al., 2019; Pan et al., 2021; Tang et al., 2020) or construct spatio-temporal graphs of actors and objects (Arnab et al., 2021b; Pan et al., 2021; Sun et al., 2018; Zhang et al., 2019). In contrast, we show how we can achieve similar accuracy improvements by using a simple model, and altering the training strategy. By leveraging our proposed co-finetuning strategy with additional, public clip-classification datasets and large-scale pretraining we achieved substantial improvements (up to 3.4 mAP points or 14.5% relative) on the challenging AVA and AVA-Kinetics datasets. In particular, our co-finetuning method improved substantially on the rare classes (up to 46.4% relative improvement) in these long-tailed datasets, as it had a regularising effect, enabling the network to learn feature representations that transfer between different datasets.

Future work is to explore co-finetuning for other tasks and domains, and also to use more complex models.

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

Table 6: Comparison of traditional finetuning, co-pretraining and co-finetuning. "Total Iterations" denotes the total training iterations of all stages, in thousands, using a batch size of 128. We follow the same settings as Table 2 of the paper.

| Pretrain | Finetune | AVA | K400 | MiT | Total Iterations (k) |
|---|---|---|---|---|---|
| *Traditional transfer* | | | | | |
| K400 | MiT | – | – | 37.2 | 178.8 |
| MiT | K400 | – | 76.3 | – | 178.8 |
| MiT→K400 | AVA | 25.1 | – | – | 211.8 |
| *Co-pretraining* | | | | | |
| MiT + K400 | – | – | **76.7** | 38.5 | 178.8 |
| MiT + K400 | AVA | 25.5 | – | – | 211.8 |
| *Co-finetuning* | | | | | |
| – | MiT + K400 + AVA | **26.3** | **76.7** | **38.9** | 211.8 |

## Appendix

In this appendix, we include our broader impact statement (Sec. A), additional ablation experiments (Sec. B), and reproducibility statement and additional experimental details (Sec. C).

## A   Broader impact

Our work presents an improved method for performing detection and classification tasks in video. These video recognition tasks represent a general technology with a wide range of potential applications. While we are unaware of all potential applications, it is important to be aware that each application has its own merits and societal implications depending on how the system is built and deployed. We also note that training datasets may contain biases that may render models trained on them unsuitable for certain applications.

## B   Additional experiments

**Co-finetuning compared to training for more iterations**   In this experiment, we confirm whether our improvements from co-finetuning are just because we are actually training for longer:

We finetuned our model for AVA for 53 epochs, which is the same number of iterations as co-finetuning on AVA for 20 epochs and Kinetics 400 for 30 epochs. Traditional finetuning gets only 23.8 mAP, whilst co-finetuning achieves 25.2 mAP (Tab. 1a), showing that our improvements are from our proposed method, not training longer.

**Jointly training on upstream datasets**   Instead of "co-finetuning", it is also possible to train jointly on multiple upstream datasets, and then finetune on a single downstream dataset. We name this configuration "Co-pretraining", as shown in Tab. 6.

We observe that "co-pretraining" does still provide benefits in terms of accuracy on both upstream and downstream datasets. However, training jointly on both upstream and downstream datasets, as done in the experiments in the main paper, perform the best.

**Analysis of regularisation**   The main paper hypothesised that co-finetuning improves overall accuracy as it has a regularising effect. To investigate this further, we follow the definition of regularisation as "any modification we make to a learning algorithm that is intended to reduce its generalisation error but not its training error" Goodfellow et al. (2016). Therefore, to validate that our proposed co-finetuning has a regularisation effect, we compare the difference in the validation and training losses (ie the generalisation error) between the baseline and our proposed co-finetuning method.

Table 7: Analysis of regularisation due to co-finetuning. We compare the generalisation error (difference between training and validation errors) across the datasets that we co-finetune on. We observe that co-finetuning has a regularisation effect in that it decreases the generalisation error across all three datasets.

(a) Moments in Time

| Model | Training error | Validation error | Generalisation error |
|---|---|---|---|
| Baseline | 2.14 | 3.15 | 1.01 |
| Ours, co-finetuned | 2.17 | 2.74 | **0.57** |

(b) Kinetics 400

| Model | Training error | Validation error | Generalisation error |
|---|---|---|---|
| Baseline | 1.24 | 1.53 | 0.29 |
| Ours, co-finetuned | 0.90 | 0.93 | **0.03** |

(c) AVA

| Model | Training error | Validation error | Generalisation error |
|---|---|---|---|
| Baseline | 2.59 | 6.77 | 4.18 |
| Ours, co-finetuned | 2.66 | 5.78 | **3.12** |

Table 8: Effect of spatial resolution on performance on AVA, measured by the mAP. We denote the resolution as height × width, where the height is the shorter side typically reported in the literature. Observe how increasing the spatial resolution consistently improves results for both the baseline and our co-finetuning method. Moreover, note how the improvements achieved by co-finetuning are consistent across all of these resolutions, using a ViViT-Base backbone.

| Resolution | Baseline | Co-finetuning |
|---|---|---|
| 140 × 224 | 23.4 | 26.3 |
| 220 × 352 | 26.9 | 29.1 |
| 260 × 416 | 27.5 | 30.1 |
| 320 × 512 | 28.5 | 31.0 |

As Tab. 7 shows, co-finetuning does indeed reduce generalisation error, as the gap between the final training and validation losses is decreased. The results below were obtained from the models in Table 2 of the main paper. Note that we used the softmax cross-entropy loss for Moments in Time and Kinetics, and the binary cross-entropy loss for AVA as it is a multilabel dataset.

**Effect of spatial resolution on performance on AVA** Table 8 studies the effect of spatial resolution on action localisation performance on AVA, using a ViViT-Base backbone. We observe consistent improvements from increasing the spatial resolution, and this is known to improve accuracy for object detection of images too (Lin et al., 2017; Singh & Davis, 2018). Moreover, the improvements achieved by co-finetuning are consistent across all the different resolutions that we evaluated.

**Effect of large-scale pre-training** As mentioned in the main paper, the ViViT model (Arnab et al., 2021a) that we use as our backbone is initialised from a ViT image model (Dosovitskiy et al., 2021) that is trained on ImageNet-21K (Deng et al., 2009), like other popular video architectures (Bertasius et al., 2021; Liu et al., 2022; Yan et al., 2022; Zhang et al., 2021).

In Tab. 9, we also consider initial pretraining on the larger Weak Textual Supervision (WTS) (Stroud et al., 2020) dataset which consists of about 60 million weakly labelled videos scraped from the web. Table 9 shows that co-finetuning (with K400, MiT and SSv2) still improves our model's accuracy in this case by 1.9 points. Note that these models are trained with a resolution of 320 on the shorter side.

**Per-class results** Figure 3 shows detailed per-class results of our best model on the AVA dataset.

Table 9: Effect of using the web-crawled WTS dataset (Stroud et al., 2020) for initial pre-training, using a ViViT-B 320p model. Even with stronger initialisation, co-finetuning provides benefits.

|  | AVA |
| --- | --- |
| Traditional finetuning | 33.3 |
| Co-finetuning | **35.2** |

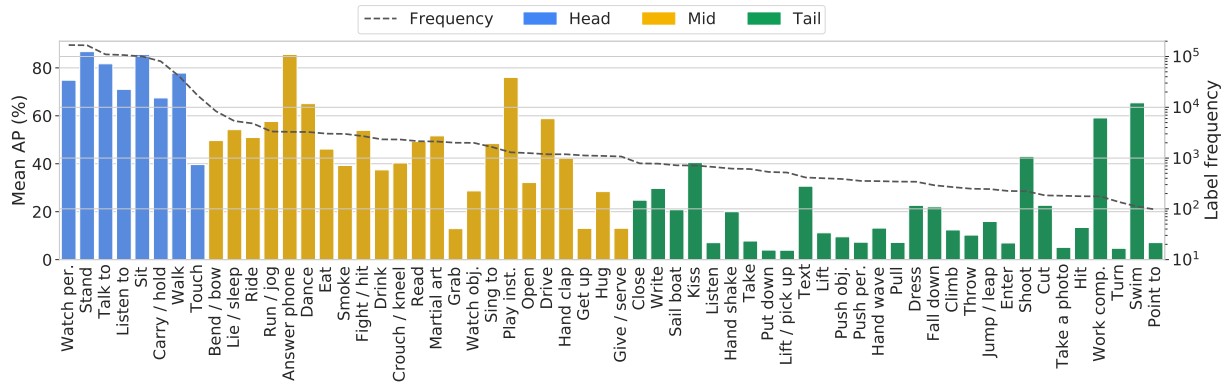

Figure 3: Per-class results on the AVA dataset (Gu et al., 2018) for our best model, achieving a mean AP of 36.1. The label frequency, shown with a log-scale, used to divide the classes into "Head", "Mid" and "Tail" categories.

## C   Reproducibility statement and additional experimental details

**Reproducibility Statement**   We will release all code and models upon acceptance. We have also included exhaustive descriptions of our implementation details below and in Sec. 4.1 of the paper. We have also followed standard experimental protocols in the community for all of the datasets that we have used for our experiments.

**Additional implementation details**   We trained all of our models with 32 frames sampled with a temporal stride of 2. We trained our models using synchronous SGD with a momentum of 0.9, and a global batch size of 128.

We set the base learning rate to 0.2 for all models, and followed a cosine learning rate schedule with a linear warm-up for the first 2.5 epochs. We trained for 20, 20, 30, 20 and 60 epochs on AVA, AVA-Kinetics, Kinetics 400, Moments in Time and Something-Something v2 respectively, for both baseline and co-finetuning strategies.

For data augmentation, we applied scale jittering to the input frames with a ratio uniformly sampled from the range $[0.65, 1.1]$. We also applied random jittering on the person detections used during training with a maximum perturbation ratio of 0.15. For additional regularisation, we also applied label smoothing (Szegedy et al., 2016) and stochastic depth (Huang et al., 2016) and their coefficients were set to 0.1 and 0.2, respectively.

For AVA, our models are trained on detected person boxes and they are first thresholded with a confidence score of 0.8 following Wu et al. (2019); Feichtenhofer et al. (2019); Arnab et al. (2021b); Fan et al. (2021). For AVA-Kinetics, ground-truth person detections are used as the samples for training and we keep detections with a confidence score greater than 0.2 for evaluation. The differences between the confidence score thresholds is because for AVA, we used the Faster-RCNN(Ren et al., 2015) detector originally trained by Wu

et al. (2019), and for AVA-Kinetics, we used the CentreNet (Zhou et al., 2019) detector originally trained by Li et al. (2020).

**List of head, mid and tail classes**  Table 10 lists the "Head", "Mid" and "Tail" classes based on their label frequencies in the AVA v2.2 (Gu et al., 2018) training set.

We defined "Head" classes as having more than 10,000 labelled examples, and "Tail" classes have less than 1,000 examples. Classes not falling into either of these categories are labelled "Mid".

Table 10: Label frequencies of each class in the AVA v2.2 (Gu et al., 2018) training set. We define "Head" classes as having more than 10,000 labelled examples, and "Tail" classes having less than 1,000 examples. Classes which do not fall into either category are "Mid" classes.

| *Head* (8 classes) | | | | | |
|---|---|---|---|---|---|
| watch (a person) | 168148 | stand | 166357 | talk to (e.g., self, a person, a group) | 110267 |
| listen to (a person) | 106816 | sit | 100323 | carry/hold (an object) | 80451 |
| walk | 40771 | touch (an object) | 17133 | | |

| *Mid* (23 classes) | | | | | |
|---|---|---|---|---|---|
| bend/bow (at the waist) | 8349 | lie/sleep | 5356 | ride (e.g., a bike, a car, a horse) | 4808 |
| run/jog | 3337 | answer phone | 3279 | dance | 3267 |
| eat | 3025 | smoke | 2991 | fight/hit (a person) | 2695 |
| drink | 2335 | crouch/kneel | 2321 | read | 2146 |
| martial art | 2117 | grab (a person) | 2003 | watch (e.g., TV) | 1993 |
| sing to (e.g., self, a person, a group) | 1643 | play musical instrument | 1297 | open (e.g., a window, a car door) | 1251 |
| drive (e.g., a car, a truck) | 1188 | hand clap | 1187 | get up | 1124 |
| hug (a person) | 1103 | give/serve (an object) to (a person) | 1073 | | |

| *Tail* (29 classes) | | | | | |
|---|---|---|---|---|---|
| close (e.g., a door, a box) | 786 | write | 777 | sail boat | 719 |
| kiss (a person) | 714 | listen (e.g., to music) | 668 | hand shake | 619 |
| take (an object) from (a person) | 608 | put down | 533 | lift/pick up | 519 |
| text on/look at a cellphone | 415 | lift (a person) | 400 | push (an object) | 387 |
| push (another person) | 355 | hand wave | 351 | pull (an object) | 344 |
| dress/put on clothing | 342 | fall down | 291 | climb (e.g., a mountain) | 268 |
| throw | 249 | jump/leap | 245 | enter | 225 |
| shoot | 222 | cut | 184 | take a photo | 181 |
| hit (an object) | 177 | work on a computer | 176 | turn (e.g., a screwdriver) | 137 |
| swim | 111 | point to (an object) | 97 | | |

