# OpenReview forum: "Beyond Traditional Transfer Learning: Co-finetuning for Action Localisation"
_TMLR — Rejected by TMLR_

### Review · Reviewer_xU9u · 2023-01-21

**Summary Of Contributions:**

This paper proposes an alternative to the widely used pretrain-and-finetune paradigm for spatial temporal action localization. The authors propose to co-finetune on large upstream datasets and small downstream datasets at the same time.

**Audience:**

Yes

**Broader Impact Concerns:**

No other concerns

**Claims And Evidence:**

No

**Requested Changes:**

Please refer to weaknesses.

**Strengths And Weaknesses:**

Strengths:

The paper is well written and easy to understand and the proposed method could achieve better performance.

Weaknesses:

1. The motivation of co-finetuning is questionable. Traditionally, researchers choose to use the pretrain-and-finetune paradigm not because it is the best way to achieve good performance. Instead, it is because this paradigm is efficient and friendly. In the case of spatial temporal action localization, researchers set the benchmark with the pretrain-and-finetune paradigm because this makes it more accessible to researchers that can not afford training models on the large-scale Kinetic dataset. Therefore, the motivation of co-finetuning does not come from a very meaningful previously unnoticed research gap.

2. The authors state that co-finetuning performs better than traditional transfer learning when using the same total amount of data. However, the performance should also be grounded on the same total amount of computations.

3. By co-finetuning, it is more reasonable to expect performance improvement on both upstream and downstream tasks than only on the downstream tasks. The performance improvement on upstream tasks is unclear in the paper.

---

> ### Author Response · Authors · 2023-02-12
> **Rebuttal**
>
> Thank you for your considered review. Our responses are below:
>
> ### The motivation of co-finetuning is questionable
>
> Thank you. There are numerous benefits of our co-finetuning strategy
> - Improved accuracy on “upstream” and “downstream” tasks, as illustrated in Tables 1 and 2.
> - A single model that is capable of performing multiple tasks, and improves upon all of them, as shown in Tables 1 and 2.
> - When we have multiple upstream datasets, it is not clear how to leverage them with traditional transfer learning. However, as shown in Tables 2 and 3, our proposed co-finetuning achieves further improvements in this setting.
> - Improving accuracy more on rare classes, as shown in Table 3.
> - Empirically being able to outperform intricate, state-of-the-art spatio-temporal action localisation models by using only a simple baseline model, and improving the training strategy instead, as illustrated in Table 5.
> - We achieve these empirical improvements with our proposed co-finetuning without increasing the total training time (please also see point below).
>
> Moreover, as we are not aware of previous work training models in this way in the computer vision community, we believe that our study can provide value to the community.
>
>
> ### Performance should be grounded on total amount of computation
>
> Thank you. The total computational cost of co-finetuning is actually identical to traditional pretraining and finetuning. This is because we train for the same number of epochs on the upstream datasets. Therefore, all experiments (baselines and our method) in Tables 1a, 1b and 2 have the same training time.
> Therefore, our co-finetuning strategy improves results over traditional finetuning when using both the same amount of training data and computation.
> We have updated Tables 1a, 1b and 2 in the revision. Following TMLR guidelines, we will upload the revision once the final review has been received. In the meantime, we have also included the updated tables below for easier reference.
>
> Table 1: Comparison of traditional finetuning, and our proposed co-finetuning, with the same total amount of data. Observe how co-finetuning consistently outperforms traditional finetuning. We report single-view accuracies for all datasets, using a ViViT-Base backbone. We report the total number of training iterations, in thousands, using a batch size of 128.
>
> a) Kinetics 400 as upstream classification dataset.
> | Pretrain |  Finetune  |  AVA | K400 | Total training iterations (k) |
> |:--------:|:----------:|:----:|:----:|:-----------------------------:|
> |   K400   |     --     |  --  | 74.9 |              55.2             |
> |   K400   |     AVA    | 23.4 |  --  |              88.2             |
> ||||||
> |    --    | K400 + AVA | **25.2** | **76.2** |              88.2             |
>
> b) Moments in Time as upstream classification dataset.
> | Pretrain |  Finetune  |  AVA | MiT | Total training iterations (k) |
> |:--------:|:----------:|:----:|:----:|:-----------------------------:|
> |   MiT   |     --     |  --  | 36.8 |              123.5             |
> |   MiT   |     AVA    | 24.8 |  --  |              156.5             |
> ||||||
> |    --    | MiT + AVA | **26.1** | **38.1** |              156.5             |
>
>
> Table 2: Comparison of traditional finetuning, and our proposed co-finetuning, using two upstream datasets. Co-finetuning consistently outperforms traditional finetuning with the same overall data, even considering that there are multiple orders in which two upstream datasets can be pretrained in with traditional finetuning (K400→MiT denotes first pretraining on K400 and then MiT). The final column shows the total number of training iterations, in thousands, using a batch size of 128.
> |   Pretrain  |     Finetune     |    AVA   |   K400   |    MiT   | Total training iterations (k) |
> |:-----------:|:----------------:|:--------:|:--------:|:--------:|:-----------------------------:|
> |     K400    |        MiT       |    --    |    --    |   37.2   |             178.8             |
> |     MiT     |       K400       |    --    |   76.3   |    --    |             178.9             |
> | K400 -> MiT |        AVA       |   25.3   |    --    |    --    |             211.8             |
> | MiT -> K400 |        AVA       |   25.1   |    --    |    --    |             211.8             |
> |||||||
> |             | MiT + K400 + AVA | **26.3** | **76.7** | **38.9** |             211.8             |
>
>
> ### “By co-finetuning, it is more reasonable to expect performance improvement on both upstream and downstream tasks than only on the downstream tasks”
>
> Please can the reviewer clarify this comment? In our experiments, video classification datasets such as Kinetics and Moments in Time are considered the upstream tasks, and action localisation on AVA was the downstream task.
>
> As shown in Tables 1a, 1b and 2, we improved upon all of these tasks. Therefore, we did in fact observe performance improvements in upstream and downstream tasks like the reviewer is asking for.

---

### Review · Reviewer_N47V · 2023-01-29

**Summary Of Contributions:**

This paper proposes a new transfer learning method called co-finetuning and confirms the effectiveness of the proposed method through its application to an action localization task. In co-finetuning, the finetuning of a model is performed simultaneously using both upstream and downstream data sets. The authors claim that this learning strategy provides a regularization effect, especially for rare classes in the target dataset for which the network may be overfitting. Experiments on AVA and AVA-Kinetics datasets show improved classification performance for rare classes in long-tail datasets.


**Audience:**

Yes

**Broader Impact Concerns:**

nothing

**Claims And Evidence:**

Yes

**Requested Changes:**

- It seems to be essentially the same as the normal multitask learning framework, as mentioned in the WEAKNESSES section. Is it possible to demonstrate theoretically the essential difference from ordinary multitask learning?

- There are other similar problem settings, such as multi-source domain adaptation, besides the proposed method and multitask learning. Their relevance and differences should be further discussed.

- The proposed method learns the effects of regularization and discriminative features, according to the authors. The paper only discusses these by improving the final classification score. Could visualization of the feature distribution or other means demonstrate regularization or discriminative feature learning more directly?

- The proposed method does not address any problems specific to action localization other than the pipeline it is applied to. Beyond the action localization task, can the authors discuss its generality?

- There are some exaggerated expressions like "significant" improvement etc. in the description of the paper. Correcting them to more objective terms would be better.


**Strengths And Weaknesses:**

- Strengths

-- The proposed method is a very simple learning method for finetuning models using upstream and downstream datasets simultaneously.

-- In this paper, several action recognition datasets such as AVA, Kinetics, Moments in Time, and Something-Something are used in combination to compare the transfering performance of the proposed method with the usual finetuning method.

-- The experiments show that the co-finetuning method outperforms the usual finetuning method for all dataset combinations using the above action recognition datasets.

-- The datasets are divided into tail, mid, and head according to the number of data contained in the action category. The paper analyzes how much the transfering performance is improved in each category.

-- The paper is easy to read. The proposed method is easily understood.

- Weaknesses

-- Compared to other previously proposed methods, such as multi-task learning, the proposed method does not have many significant differences.

-- The proposed learning method seems to be general. However, it is only applied to one backbone model or pipeline, and its usefulness and generality in other pipelines is unclear.

-- Problems specific to action localization task do not seem to be addressed by the proposed method. There is little relationship between the proposal and action localization methods.

---

### Review · Reviewer_u1GF · 2023-03-01

**Summary Of Contributions:**

This paper introduces a new approach named co-finetuning: finetuning a single imagenet-pretrained model on multiple upstream and downstream tasks.

Advantages of co-finetuning:
* Co-finetuning outperforms traditional transfer learning with the same total amount of data.
* Co-finetuning on multiple _upstream_ datasets further improves performance.

The paper provides fair amount of experiments to show the advantages of co-finetuning are valid in multiple daataset settings.

**Audience:**

No

**Broader Impact Concerns:**

Sufficient.

**Claims And Evidence:**

Yes

**Requested Changes:**

Please run proofreading to fix grammar and expressions.

Please address the weaknesses and questions.

**Strengths And Weaknesses:**

## Strengths

Co-finetuning is an interesting approach.

----------

## Weakness

The wording "co-finetuning" is somewhat misleading as it does not include pretraining except ImageNet initialization.

Comparisons between `co-finetuning from ImageNet initialization` and  `pretraining on upstream dataset and co-finetuning` is missing. For example, would pretraining on K400 or MiT and co-finetuning on K400 + AVA outperform co-finetuning on K400 + AVA from ImageNet initialization?

Experiments do not provide how long each configuration takes to train. Readers might be interested in comparing costs for the performance boost.

Performance over iterations?

----------

## Questions
Is co-finetuning a new approach? The continual learning community considers pretraining + co-finetuning as the upper bound of pretraining + finetuning.

What problem does co-finetuning solve? "Co-finetuning outperforms other settings" is not enough message for a publication.

Would it work on different tasks other than spatio-temporal action detection?

Limiting the statements on performance within a condition of "the same total amount of data" discomforts me. Are there cases where traditional transfer learning may use more data than co-finetuning or vice versa?

----------

## Minor

Please provide the definition of person-proposal $y_j$. Maybe (x, y, w, h)?

Duplicate notation: person proposal, predicted logits, and GT label are denoted as $y$.

---

### Decision · Action_Editors · 2023-04-22

**Recommendation:** Reject

**Comment:**

The reviewers acknowledged that the proposed approach is simple and effective on action recognition and localization datasets. They also mentioned that the paper is well written and easy to read.

However, there was a unanimous concern about the proposed co-finetuning strategy. Specifically, the reviewers found the motivation of co-finetuning to be unclear and the claimed technical contribution around co-finetuning to be questionable. Specifically, they called out that co-finetuning can be viewed as the conventional multi-task learning, and therefore, the motivation of co-finetuning did not come from a meaningful previously unnoticed research gap. In the authors' rebuttal, they argued that while the notion of co-finetuning may appear indistinguishable from multi-task learning in certain fields such as NLP, it is not the case in the computer vision community. Unfortunately, this argument isn't correct (see omnivore, tensor2tensor, unified-io, etc.)

Another major concern was this paper's focus on specific application domain of action recognition and localization, when the notion of co-finetuning is much general to many different tasks. The reviewers expressed concerns that empirical evidence on the limited evaluation scenario does not provide generalizable insights to the wider TMLR community.

Due to these reasons, all three reviewers recommended reject recommendation. This action editor have carefully reviewed the submission, the reviews, and the rebuttal, and did not find strong reasons to overturn their recommendations. We therefore recommend reject at this time.

**Audience:**

TMLR's audience on video action localization might find this paper relevant to their interests.

**Claims And Evidence:**

There was a unanimous concern about the claimed novelty of co-finetuning: _"motivation of co-finetuning is questionable"_ (``xU9u``), _"the proposed method does not have significant differences to multi-task learning"_ (``N47V``), _"the wording co-finetuning is somewhat misleading as it does not include pretraining except ImageNet initialization"_ (``u1GF``). Summarizing the concerns, there is overall an over-claim issue because of the strong resemblance of co-finetuning with multi-task learning.

The authors rebutted that, although co-finetuning is identical to multi-task learning in NLP, it is considered novel in computer vision because multi-task learning has been used in a setting where a model predicts multiple outputs from the same input. However, this argument isn't entirely correct -- multi-task learning with multiple inputs and outputs has been considered in CV community (e.g., omnivore, tensor2tensor, unified-io, and so on).

There was a related concern that, although the notion of co-finetuning is general, it is demonstrated only on action localization. It is worth mentioning that the writing certainly alludes to the generality of the approach (notice in the abstract it is only at the very last sentence where they mention action localization).

Given these concerns, the paper fails to provide strong evidence that supports the claimed novelty of co-finetuning.